# Housing during the asylum process and its association with healthcare utilization for common mental disorders among refugees in Sweden: A nationwide cohort study

**Charlotta van Eggermont Arwidson**[1,2]*, **Jessica Holmgren**[3], **Kristina Gottberg**[1], **Petter Tinghög**[2,4]

**1** Division of Nursing, Department of Neurobiology, Care Sciences and Society, Karolinska Institutet, Stockholm, Sweden, **2** Department of Health Sciences, Swedish Red Cross University, Huddinge, Sweden, **3** Department of Clinical Neuroscience, Division of Psychology, Karolinska Institutet, Solna, Sweden, **4** School of Health, Care and Social Welfare, Mälardalen University, Eskilstuna, Sweden

* charlotta.arwidson@gmail.com

## Abstract

Refugees and asylum seekers face an increased risk of poor mental health, and evidence shows that housing in the post-migration context plays a crucial role in shaping their mental well-being. Research also suggests that institutional accommodations during the asylum process might be more detrimental to their mental health compared to private accommodations. We aimed to prospectively estimate the associations between housing type during the asylum process (institutional or self-organized accommodations) and healthcare utilization for common mental disorders (CMDs) after being granted a residence permit as a refugee in Sweden. This register-based cohort study includes all asylum seekers aged 18–60 who were granted residence permits in Sweden between 2010 and 2012, totaling 20,396 individuals, of whom 11,694 resided in self-organized housing (EBO) and 8,702 in accommodation centers (ABO). Using a generalized estimating equation (GEE), we estimated the associations between housing type (ABO or EBO) and prescriptions for antidepressants or anxiolytic medication, as well as specialized in- and outpatient visits with a diagnosis of CMDs, over a five-year follow-up period after being granted a residence permit. The adjusted odds ratio (controlled for sociodemographic factors) showed that those who had lived in ABO, compared with EBO, had a greater risk of any antidepressant or anxiolytic prescriptions (OR = 1.32, [1.21–1.44]) as well as any specialized in- or outpatient visits with a CMD diagnosis (OR = 1.41 [1.27–1.52]). Our results demonstrate that former asylum seekers who have lived in institutional housing use more mental healthcare services than those who have lived in self-organized housing, even when potential sociodemographic confounders and mediators are adjusted for. These associations persisted for up to five years after they had received a residence permit,

**Data availability statement:** The registry data used in this study contains pseudonymized patient-level, medically, sensitive information from nationwide registers. To safeguard the personal integrity of the participating individuals, and following Swedish law—specifically, the Public Access to Information and Secrecy Act—as well as the terms of our ethical approval, such data cannot be made publicly available. Register data can only be made available to someone else after a formal request to access the data. The public rules on publicity and confidentiality necessitate a data-protection assessment, which includes, among other things, that the authority responsible for the research performs a new risk assessment before the data can be disclosed. These rules also apply to requests for data access from researchers not involved in our project, within our organization. All data will be securely stored on a protected server in de-identified form and retained in accordance with applicable laws and ethical guidelines. Access will be restricted to authorized research personnel involved in the research project, and strict security measures will be in place. Data will be preserved for long-term use within the scope of the study and may be archived or securely deleted as per institutional policies. Although the authors cannot make their study's data publicly available at the time of publication, all authors commit to make the data underlying the findings described in this study fully available without restriction to those who request the data, in compliance with the PLOS Data Availability policy. For data sets involving personally identifiable information or other sensitive data, data sharing is contingent on the data being handled appropriately by the data requester and in accordance with all applicable local requirements. However, the statistical code used for the analyses is available in the Supporting Information file S1 Data. For further questions regarding data availability, contact: Charlotta van Eggermont Arwidson (author), charlotta.arwidson@ki.se Kristina Gottberg (author), kristina.gottberg@ki.se Petter Tinghög (author), petter.tinghog@ki.se Richard Bränström, (Associate Professor, Karolinska Institutet, institutional point of contact and data custodian), richard.branstrom@ki.se

**Funding:** The author(s) received no specific funding for this work.

**Competing interests:** The authors have declared that no competing interests exist.

highlighting that when asylum seekers live in institutional housing it is especially important to discuss how health can be promoted during the asylum-seeking period.

## Introduction

Research has consistently shown that refugees and asylum seekers are at a heightened risk of developing mental ill-health, especially common mental disorders (CMDs) such as depression and anxiety disorders [1,2]. Additionally, evidence indicates that post-migration living conditions significantly increase vulnerability to mental health concerns among these groups [3,4].

In recent decades, there has been growing scientific interest in the relationship between housing and health among asylum seekers and refugees, emphasizing that housing conditions in resettlement countries have a significant impact on health [5–7]. Various health risks exist: For instance, physical health risks involve structural factors such as substandard housing or overcrowding. The COVID-19 pandemic, for example, clearly illustrated how living conditions at collective accommodation centers increased asylum seekers' risk of being infected [8,9]. Studies also highlight various pathways connecting housing to mental health for asylum seekers and refugees, including factors like insecure tenure, frequent relocations, and the need to share housing due to financial constraints [6]. Other psychosocial factors shown to link housing with mental health included lack of privacy, lack of safety, unfamiliarity or feeling unsafe in neighborhood/location, institutional practices, dissatisfaction with the home, and lack of control [10].

There is also a growing body of research showing how different asylum support systems, including different types of accommodations, influence refugees' health and integration [11]. In this line of research there has been a specific focus on the distinctions between private and institutional accommodations, such as collective housing. For instance, in two studies from Germany the researchers concluded that residing in collective and state-provided accommodations as opposed to private ones was linked to increased distress and decreased life satisfaction [12,13]. However, as national asylum systems sometimes differ substantially in their policies and practices, leading to diverse living conditions for asylum seekers, it is important to acknowledge that research findings may not be directly transferable from one context to another. In Sweden, for example, since the beginning of the 90s there have essentially been two options for asylum seekers when it comes to accommodations. To those in need (approx. 40% of all asylum seekers), the Swedish Migration Agency offers housing on a no-choice and dispersal basis (institutional housing, Swedish acronym ABO) [14]. State-provided housing is generally evenly distributed between collective housing in large accommodation centers (also called corridor housing) and in apartment housing (apartments located in the community, usually reserved for families) [15]. The other option for an asylum seeker is to arrange their own housing (approx. 60% do this), most commonly with friends or family already living in Sweden, on a daily-allowance-only basis (own housing, Swedish acronym EBO).

In the Swedish context, several studies have explored CMDs among different refugee groups, identifying various risk factors such as gender, country of origin, and education [16,17]. However, no study has specifically examined housing as a risk factor. In terms of studies examining asylum seekers and their housing, several studies have shed light on the conditions within ABO, particularly the general living conditions and residents' mental well-being [18–20]. Conversely, little attention has been given to the living circumstances of asylum seekers who opt for EBO, and the area thus remains relatively underexplored in research. One qualitative study exploring the experiences of asylum seekers in EBO, the only one of its kind, described crowded living conditions and frequent relocation. However, it also found an overall experience of autonomy, a feeling of reduced supervision in self-organized accommodations and resistance to dispersal or assigned accommodations [21]. In addition, a governmental report from 2008 examining the impacts of EBO on housing and labor market integration concluded that refugees who had resided in EBO during their asylum period experienced slightly improved employment and housing integration compared to those who had been placed in ABO after being granted a residence permit [22]. However, aside from studies comparing ABO and EBO concerning housing integration and employment, there are currently no studies in the Swedish context that specifically compare different housing types during the asylum period and their relation to different health outcomes. Hence, there remains a significant gap in knowledge regarding this issue.

It is important to deepen the knowledge about how housing affects health through this type of comparison, and it is also relevant from a societal perspective. There is a general trend at both the European level and in Sweden towards accommodating asylum seekers in large, camp-like structures [23]. In Sweden, the EBO regulation (allowing self-organized housing) will be abolished in 2026 and asylum seekers will primarily reside in institutionalized and centralized accommodation centers throughout the asylum process [24]. The political debate primarily focuses on how EBO negatively impacts asylum seekers, particularly through issues like overcrowding, frequent relocations and segregation. However, given the scarcity of research from the Swedish context there is still uncertainty as to the impact of various housing situations on asylum seekers' mental health.

Against this background, we considered the current asylum system in Sweden to provide valuable possibilities for comparison between different types of accommodations and how they relate to different mental health outcomes (i.e., mental healthcare utilization). It would also be valuable to explore this potential association in more detail to better understand how it might be mediated by sociodemographic factors and to determine whether certain subgroups of asylum seekers are more adversely affected by their housing type during the asylum process. We believe this type of research can provide valuable insights into the long-term effects of housing on asylum seekers' and refugees' mental health.

### Aim

To prospectively estimate the associations between housing type during the asylum process (institutional or self-organized accommodations) and healthcare utilization for CMDs after being granted a residence permit in Sweden. Another aim was to investigate:

- the extent to which these associations can be accounted for by potentially mediating sociodemographic factors, and

- whether the potential associations are moderated by sociodemographic factors.

## Materials and methods

### Design

This was a prospective cohort study based on longitudinal data from nationwide population registers in Sweden. The study used data from the following databases:

• the longitudinal database for integration studies (STATIV), a register maintained by Statistics Sweden, containing annually updated individual-level information on sex, age, country of origin, migration status, level of education, disposable income, geographical region, and family situation.

- the National Patient Register (NPR), maintained by the National Board of Health and Welfare, containing annually updated individual-level data on main and secondary diagnoses of mental and somatic disorders from inpatient and specialized outpatient healthcare visits (including privately run public healthcare settings but not primary care).

- the Prescribed Drug Register (PDR), also administered by the National Board of Health and Welfare, encompassing all prescription medications acquired from pharmacies in Sweden.

Information from these registers was linked using an anonymized identification number assigned to each individual by Statistics Sweden, ensuring anonymity.

## Ethics statement

This study was approved by the Regional Ethics Review Board in Stockholm (file number 2017/1736-31-01893) and conducted in accordance with Swedish regulations for registry-based research using anonymized national register data, for which individual consent is not required.

## Study population

The sample was identified from the STATIV database and included all individuals aged 18–60 who were granted a residence permit as refugees between 2010 and 2012 and who, before receiving a residence permit, were registered with the Swedish Migration Agency (SMA) as asylum seekers, e.g., had been awaiting a decision on their asylum claim in Sweden. This excluded quota refugees and individuals who received a residence permit based on family ties to a person with refugee status. The age range was determined based on criteria defining adults of working age throughout the follow-up period. These individuals were followed over a span of five years after being granted residence in Sweden. For instance, those who received a residence permit in 2010 were monitored from January 1, 2011, to December 31, 2015. Similarly, individuals who were granted a residence permit in 2011 were followed from January 1, 2012, to December 31, 2016, and so forth. The baseline data for the study population was collected from the first year following the year of receiving a residence permit, which also constitutes the first year of follow-up.

## Exposure

The exposure variable was housing type during the asylum process, as defined in the STATIV database, i.e., "From the Swedish Migration Agency with institutional housing (ABO)" and "From the Swedish Migration Agency with own accommodations (EBO)". These two codes were assigned to all former asylum seekers who, upon receiving a residence permit, were relocated from the SMA to a municipality within Sweden. Most asylum seekers typically resided in either ABO or EBO throughout the entire asylum process. However, an estimated 10% of individuals switch housing arrangements during the process—such as those residing in EBO transitioning to ABO, or vice versa, individuals living in ABO shifting to EBO (Swedish Migration Agency, personal communication, October 24, 2024). The duration of the exposure—i.e., the time it took to process an asylum case, and consequently the period an asylum seeker resided in either ABO or EBO—fluctuated based on the country and the number of appeals filed by the asylum seeker. For instance, during this period the average processing time for individuals from Somalia was 131 days, whereas those from Iraq had to wait an average of 160 days for a decision on their asylum claim [14]. The Swedish Migration Agency does not offer statistics on average processing times apart from a distribution based on applicants' citizenship. Additionally, the dataset available for this study lacks variables that would enable the calculation of individual processing times, which would allow for a comparison of length of stay based on housing type. However, there are no indications that processing times would differ, as the Swedish Migration Agency applies equal processing standards to individuals from both ABO and EBO. The overall average processing time was approximately 126 days. In institutional accommodations, the general rule is that an asylum seeker

lives in a shared room containing four to six people if single, sharing a kitchen and bathroom with several others. Families are allowed to live together in their own room or apartment.

## Potential confounders and mediators

The potential confounders and mediators (sociodemographic factors) used in this study were sex, age, region of origin, level of education, children in the household, job status, urbanicity, disposable income, and divorced/widowed. Time-dependent variables —i.e., level of education, children in the household, job status, urbanicity, disposable income, and divorced/widowed—were measured on December 31 for the baseline year and every subsequent follow-up year.

Age was categorized into the age groups 18–29 years, 30–45 years, and 46–60 years. Region of origin was primarily grouped into broader regions and continents; however, in instances in which Sweden hosts a significant population from a specific country, such as Iraq or Somalia, we have chosen to retain the categorization under the country name. This decision is based on our assessment that it does not compromise anonymity and that it can offer valuable insights in the analysis. Furthermore, to mitigate the impact of excessively small groups, which may compromise the stability of statistical estimates, South America was coded as missing (n = 20). We do not believe this has any substantive implication for the overall analysis. Hence, the five subcategories are Africa, Somalia, Asia, Iraq, and Europe. Level of education was classified based on the number of years of schooling: 0–9 years, 9–12 years, and >12 years. When information on education was missing for the baseline year or early follow-up years, information on education, when available, was retrieved from subsequent follow-up years.

The "children in the household" variable was dichotomized (yes/no), with "children" including both minor and adult (>18years) children living at home. Furthermore, a dichotomous variable (yes/no) was created to indicate job status. Urbanicity was defined based on population size, geographic density, and proximity to major cities and/or urban areas, following the definitions of the Swedish Association of Local Authorities and Regions (SALAR) [25]. Hence, geographical region was categorized into the three groups "major city", "medium-sized city", and "smaller city or rural area".

Disposable income is a variable created by Statistics Sweden that considers the total family income from all registered sources, including wages, welfare benefits, other social subsidies, and pensions, weighted according to household size and composition. It is measured as a percentage and reflects the relation to a *reasonable standard of living* (a national standard set by the Swedish government prior to the start of each calendar year [26]) e.g., 100% is equivalent with a reasonable standard of living. The variable was dichotomized into "Reasonable standard of living" (≥100%) and "Below reasonable standard of living" (<100%).

## Outcomes

The study's outcome measures were antidepressant or anxiolytic drug prescriptions as well as receiving treatment at a specialized outpatient care clinic or inpatient care visits, with diagnoses of mood or anxiety disorders registered as main or secondary diagnoses (diagnoses defined by the International Classification of Disorders version 10 (ICD-10) codes). The selected outcome measures and associated codes (ICD-10 and ATC) were intended to capture what are typically labelled common mental disorders (CMDs) [2].

Information on prescribed antidepressant or anxiolytic medication was retrieved from the Swedish Prescribed Drug Register and the outcome was dichotomized into either "no antidepressant or anxiolytic prescriptions" or "any antidepressant or anxiolytic prescriptions" following the Anatomical Therapeutic Chemical (ATC) classification system on drugs (codes N06A and N05B).

Data on main or secondary diagnoses registered when visiting specialized in- or outpatient care, coded with the ICD-10, was retrieved from the National Patient Register. The outcome was main or secondary diagnostic codes of mood disorders (codes F30–F39) or anxiety disorders (codes F40–F42). Using these codes, all individuals were categorized as

having had "no in- or outpatient healthcare visits with a CMD diagnosis" or "any in- or outpatient healthcare visits with a CMD diagnosis".

## Statistical analysis

Baseline characteristics for the ABO and EBO groups were compared using chi-square tests.

The study used a generalized estimating equation (GEE) with robust standard errors and a logit link function [27] to investigate both the relationship between the prescription of antidepressants or anxiolytics and housing type (ABO or EBO) as well as the association between specialized in- and outpatient visits with a diagnosis of CMD and housing type. The study population was followed over five years after receiving a residence permit. GEE is well suited for analyzing repeated-measures data as it accounts for the correlation between multiple within-subject measurements in longitudinal studies. Furthermore, unlike traditional methods, GEE does not require the data to follow a normal distribution and can accommodate subjects with missing values on the independent variables (both time-independent and time-dependent potential confounders and mediators) within the models. The autoregressive (AR) correlation structure was used to account for within-subject correlations in the analysis.

The results are presented as odds ratios with 95% confidence intervals (CI). In controlling for confounding, we used two different models (only variables showing statistically significant differences between ABO and EBO were included). In the first model, analyses were adjusted for sex, age, country of origin, educational level, and children in the household, potentially confounding the association between housing type and health service use. Model 2 additionally adjusted for work status, disposable income, and urbanicity. As studies have indicated [22], these factors can be influenced by housing type; for instance, whether one lives in EBO or ABO may affect employment opportunities or urbanicity. Thus, these factors may serve as mediators in the relationship between the exposure and the outcome, or, expressed differently, may lie within the causal pathway between the exposure and the outcome. The sex and country of origin variables were included in the GEE models as time-independent factors, and all other covariates were included as time-dependent variables.

Finally, to investigate potential effect modifications, interaction terms were added between sociodemographic factors and housing type. Effect modifications were evaluated using $X^2$ tests, and statistical significance was defined as $p < 0.01$ to mitigate the multiple comparison problem. Furthermore, to explore potential bias from missing values a series of sensitivity analyses were conducted, by inserting a separate code for missing values in the variables that contained missing values. This applied to the region of origin, disposable income, and level of education variables (missing values <3%). These sensitivity analyses revealed that accounting for missing values resulted in minimal changes to the associations between exposure and outcomes, i.e., no bias could be demonstrated. Additionally, sensitivity analyses were performed in which former asylum seekers from South America were included as a separate category within the 'region of origin' variable. These supplementary analyses had no impact on the primary estimates. Statistical analysis was conducted using SPSS, version 29.

## Results

The study population encompassed a total of 20,396 individuals, with 11,694 individuals registered as having resided in self-organized housing (EBO) and 8,702 as having resided in accommodation centers (ABO). Descriptive data at baseline for the two groups is shown in Table 1.

In terms of differences between the two groups regarding sex, level of education, and age there were only subtle variations. More notable differences could be seen comparing job and urbanicity, whereby having lived in EBO meant being more likely to have a job or to live in a major city. Some differences could also be noted in region of origin, whereby it was more common to have lived in EBO if the individual was from Iraq while it was more common to have lived in ABO if one was from Africa (except Somalia). Those having lived in ABO were also more likely to cohabitate with children and to have a reasonable living standard. When it came to events of divorce or of being widowed, no differences could be detected between the groups (Table 1).

**Table 1. Sociodemographic characteristics of study population (n = 20,396).**

**Baseline**

| | ABO (%) | EBO (%) | X2 (p-value) |
|---|---|---|---|
| *Sex* | | | df1, 6.6 (<0.01) |
| Man | 4716 (54.2) | 6549 (56.0) | |
| Woman | 3986 (45.8) | 5145 (44.0) | |
| *Age group* | | | df2, 34.36 (<0.01) |
| 18–29 | 4159 (47.8) | 5344 (45.7) | |
| 30–45 | 3637 (41.8) | 4823 (41.2) | |
| 46–60 | 906 (10.4) | 1527 (13.1) | |
| *Region of origin* | | | df4, 1127.7 (<0.01) |
| Africa* | 1956 (23.2) | 1196 (10.6) | |
| Somalia | 2633 (31.2) | 3965 (35.2) | |
| Asia** | 3206 (37.9) | 4206 (37.4) | |
| Iraq | 296 (3.5) | 1613 (14.3) | |
| Europe | 358 (4.2) | 279 (2.5) | |
| *Job* | | | df1, 282.7 (<0.01) |
| Yes | 572 (6.6) | 1633 (14) | |
| No | 8130 (93.4) | 10061 (86) | |
| *Level of education* | | | df2, 28.44 (<0.01) |
| 0–9 years | 4734 (54.4) | 6274 (53.7) | |
| 10–12 years | 2038 (23.4) | 2484 (21.2) | |
| >12 years | 1736 (19.9) | 2652 (22.7) | |
| *Urbanicity* | | | df2, 1967.1 (<0.01) |
| Major city | 1238 (14.2) | 4546 (38.9) | |
| Medium-sized city | 4406 (50.6) | 5417 (46.3) | |
| Smaller city or rural area | 3058 (35.1) | 1731 (14.8) | |
| *Disposable income* | | | df1, 96.75, (<0.01) |
| Reasonable living standard (>100%) | 5001 (57.7) | 5844 (50.7) | |
| Below reasonable living standard (<100%) | 3670 (42.3) | 5682 (49.3) | |
| *Children in the household* | | | df1, 80.40 (<0.01) |
| Yes | 3574 (41.1) | 4084 (34.9) | |
| No | 5128 (58.9) | 7610 (65.1) | |
| *Divorced/widowed*** | 424 (4.9) | 570 (4.9) | df1, <0.01 (0.99) |

*Not Somalia,

**Not Iraq,

***Individuals with at least one event over the five-year follow-up period

In the first GEE model, adjusting for sex, age, region of origin, educational level, and children in the household, it is shown that those who had lived in ABO had a greater risk of any antidepressant or anxiolytic prescriptions (OR = 1.32, [1.21–1.44]) as well as any specialized in- or outpatient visits with a CMD diagnosis (OR = 1.41 [1.27–1.52]), compared with those who had lived in EBO (Table 2).

In addition, the analysis showed that being a woman, having a higher age, coming from either Asia, Iraq or Europe (compared to coming from Africa), having a lower level of education, and not cohabitating with children carried higher risks for both any antidepressant or anxiolytic prescriptions and any in- or outpatient visits with a CMD diagnosis.

PLOS Global Public Health

**Table 2. General estimating equation models for estimating the associations between sociodemographic factors and any antidepressant or anxiolytic prescriptions and any specialized in- or outpatient visits with a CMD diagnosis.**

| Variables | Any antidepressant or anxiolytic prescriptions | | | | Any in- or outpatient healthcare visits with a CMD diagnosis | | | |
|---|---|---|---|---|---|---|---|---|
| | Model 1 | | Model 2 | | Model 1 | | Model 2 | |
| | OR | CI | OR | CI | OR | CI | Or | CI |
| **Housing type** | | | | | | | | |
| Assigned housing (ABO) | 1.32 | 1.21–1.44 | 1.39 | 1.27–1.52 | 1.41 | 1.20–1.65 | 1.43 | 1.22–1.69 |
| Self-organized housing (EBO) | 1 | | 1 | | 1 | | 1 | |
| **Sex** | | | | | | | | |
| Man | 1 | | 1 | | 1 | | 1 | |
| Woman | 1.59 | 1.46–1.73 | 1.48 | 1.36–1.61 | 1.72 | 1.48–2.01 | 1.57 | 1.35–1.83 |
| **Age group** | | | | | | | | |
| 18–29 | 1 | | 1 | | 1 | | 1 | |
| 30–45 | 1.69 | 1.55–1.85 | 1.73 | 1.59–1.89 | 1.22 | 1.05–1.42 | 1.26 | 1.08–1.47 |
| 46–65 | 2.83 | 2.54–3.16 | 2.83 | 2.54–3.15 | 1.49 | 1.22–1.82 | 1.46 | 1.19–1.78 |
| **Region of origin** | | | | | | | | |
| Africa* | 1 | | 1 | | 1 | | 1 | |
| Somalia | 1.17 | 0.99–1.38 | 1.17 | 0.98–1.38 | 0.68 | 0.49–0.95 | 0.67 | 0.48–0.92 |
| Asia** | 3.70 | 3.18–4.32 | 3.71 | 3.18–4.33 | 4.04 | 3.13–5.23 | 4.02 | 3.11–5.20 |
| Iraq | 3.94 | 3.28–4.75 | 3.97 | 3.32–4.77 | 4.29 | 3.12–5.89 | 4.17 | 3.03–5.72 |
| Europe | 4.93 | 3.90–6.22 | 5.23 | 4.16–6.60 | 6.90 | 4.84–9.82 | 7.21 | 5.07–10.27 |
| **Level of education** | | | | | | | | |
| 0–9 years | 1.24 | 1.12–1.38 | 1.23 | 1.11–1.37 | 1.35 | 1.13–1.63 | 1.34 | 1.11–1.62 |
| 10–12 years | 1.06 | 0.95–1.18 | 1.09 | 0.97–1.21 | 1.08 | 0.89–1.32 | 1.13 | 0.93–1.38 |
| >12 years | 1 | | 1 | | 1 | | 1 | |
| **Children in the household** | | | | | | | | |
| Yes | 1 | | 1 | | 1 | | 1 | |
| No | 1.39 | 1.29–1.55 | 1.40 | 1.29–1.51 | 1.81 | 1.57–2.08 | 1.87 | 1.61–2.16 |
| **Job** | | | | | | | | |
| Yes | | | 1 | | | | 1 | |
| No | | | 1.67 | 1.55–1.79 | | | 2.06 | 1.76–2.31 |
| **Disposable income** | | | | | | | | |
| Reasonable living standard (≥100%) | | | 1.08 | 1.01–1.14 | | | 1.13 | 1.01–1.27 |
| Below reasonable living standard (< 100%) | | | 1 | | | | 1 | |
| **Urbanicity** | | | | | | | | |
| Major city | | | 1.37 | 1.23-1.53 | | | 1.26 | 1.02–1.54 |
| Medium-sized city | | | 0.98 | 0.89–1.09 | | | 0.99 | 0.82–1.19 |
| Smaller city or rural area | | | 1 | | | | 1 | |

Model 1: Adjusted for sex, region of origin, age, level of education, children in the household (first two variables measured at baseline, all others measured at every follow-up year). Model 2: Model 1 + job, disposable income, urbanicity (all measured at every follow-up year).

*Not Somalia,

**Not Iraq.

In Model 2, controlling for potentially mediating factors, a similar pattern was observed, indicating that the introduced factors had a minor influence on the associations between housing type and antidepressant or anxiolytic prescription (OR = 1.39 [1.27–1.52]) or specialized in- or outpatient visits with a CMD diagnosis (OR = 1.43 [1.22–1.69]. Additionally, the Model 2 analysis showed that having no job and living in a major city carried higher risks for both any antidepressant or

PLOS Global Public Health

**Table 3. Odds ratio and 95% confidence interval for any antidepressant or anxiolytic prescriptions and any specialized in-or outpatient visits with a CMD diagnosis in any of the follow-up years for people from ABO compared to those from EBO.**

| Follow-up year | Any antidepressant or anxiolytic prescriptions | | | | Any in- or outpatient healthcare visits with a CMD diagnosis | | | |
|---|---|---|---|---|---|---|---|---|
| | Model 1 | | Model 2 | | Model 1 | | Model 2 | |
| | OR | CI | OR | CI | OR | CI | OR | CI |
| 1 | 1.34 | 1.24–1.58 | 1.45 | 1.28–1–64 | 1.43 | 1.15–1.79 | 1.43 | 1.14–1.79 |
| 2 | 1.28 | 1.13–1.44 | 1.33 | 1.18–1.51 | 1.42 | 1.14–1.77 | 1.41 | 1.12–1.77 |
| 3 | 1.22 | 1.08–1.37 | 1.27 | 1.13–1.44 | 1.57 | 1.25—1.96 | 1.57 | 1.26–1.97 |
| 4 | 1.35 | 1.2–1.52 | 1.44 | 1.28–1.63 | 1.47 | 1.17–1.86 | 1.52 | 1.2–1.92 |
| 5 | 1.31 | 1.15–1.48 | 1.4 | 1.24–1.59 | 1.24 | 0.98–1.58 | 1.29 | 1.02–1.65 |

Model 1: Adjusted for sex, region of origin, age, level of education, children in the household (first two variables measured at baseline, all others measured at every follow-up year). Model 2: Model 1+job, disposable income, urbanicity (all measured at every follow-up year).

anxiolytic prescriptions and any in- or outpatient visits with a CMD diagnosis. It was also observed that the covariates predicted the two outcomes in similar ways in the two models. Additionally, it is worth noting that the associations remain relatively stable and statistically significant at each follow-up year, reflecting the consistency of these associations throughout the entire follow-up period (Table 3).

The analysis of whether the associations between housing type and the outcomes were moderated by sociodemographic factors (Table 4) showed only small differences in how the covariates predicted the outcomes between individuals from ABO and those from EBO. Overall, this suggests that sociodemographic characteristics do not notably affect individuals from EBO in a different way than they do those from ABO in relation to the measured mental health outcomes. The only notable difference was that living in a major city appears to have a slightly greater influence on individuals who have lived in EBO, increasing the likelihood of being prescribed antidepressant or anxiolytic medication. Furthermore, individuals from EBO with a reasonable standard of living were at greater risk of having had any in- or outpatient visits. However, this scenario was not observed among individuals from ABO.

## Discussion

Our findings show that, compared to those who had lived in EBO during the asylum process, individuals from ABO ran a higher risk of both having used antidepressant or anxiolytic medication as well as receiving treatment at specialized out- or inpatient clinics with a CMD diagnosis during the five years after receiving a residence permit in Sweden. This elevated risk remained irrespective of sociodemographic factors such as age, sex, region of origin, level of education, or children in the household. We also found that the potentially mediating factors that were examined—job status, disposable income, and urbanicity—did not influence the association between housing and mental healthcare utilization in a noteworthy way; the higher risk of using mental healthcare services among people from ABO remained. In addition, the moderating effects of the tested sociodemographic variables on the relationship between housing type and outcomes were not substantial. The only notable result was that level of urbanicity appears to be a stronger predictor of being prescribed antidepressant or anxiolytic medication among individuals from EBO than those from ABO. This may be challenging to interpret, but one possible explanation is that the EBO group has been more exposed to urban environments than the ABO group. People living in EBO more often reside in larger cities during the asylum period, which increase the likelihood of encountering factors associated with a higher risk of mental disorders [28].

Caution must be observed when drawing comparisons between countries with different asylum support systems, and varying methodological approaches complicate direct comparisons. Nevertheless, we believe it is reasonable to state that our findings are consistent with previous studies in the field. First, several earlier studies have shown that living conditions

**Table 4. General estimating equation model for estimating the associations between sociodemographic factors and any antidepressant or anxiolytic prescriptions or any in- or outpatient visits with a CMD diagnosis for ABO and EBO, respectively.**

| | Any antidepressant or anxiolytic prescriptions | | | Any in- or outpatient healthcare visits with CMD diagnosis | | |
|---|---|---|---|---|---|---|
| | ABO [95% CI] | EBO [95% CI] | Effect modification $X^2$ (p-value) | ABO [95% CI] | EBO [95% CI] | Effect modification $X^2$ (p-value) |
| **Sex** | | | | | | |
| Man | 1 | 1 | df1, 1.19 (0.28) | 1 | 1 | df1, 1.58 (0.21) |
| Woman | 1.40 [1.24–1.60] | 1.54 [1.38–1.73] | | 1.42 [1.14–1.77] | 1.72 [1.39–2.12] | |
| **Age group** | | | | | | |
| 18–29 | 1 | 1 | df2, 2.15 (0.34) | 1 | 1 | df2, 7.54 (0.02) |
| 30–45 | 1.65 [1.45–1.87] | 1.81 [1.61–2.05] | | 1.10 [0.89–1.37] | 1.43 [1.15–1.77] | |
| 46–60 | 2.89 [2.45–3.42] | 2.81 [2.43–3.24] | | 1.67 [1.26–2.22] | 1.33 [1.02–1.75] | |
| **Region of origin** | | | | | | |
| Africa* | 1 | 1 | df4, 3.89 (0.42) | 1 | 1 | df4, 7.25 (0.12) |
| Somalia | 1.16 [0.93–1.46] | 1.14 [0.87–1.48] | | 0.81 [0.53–1.22] | 0.52 [0.31–0.87] | |
| Asia** | 3.77 [3.12–4.56] | 3.57 [2.78–4.59] | | 4.00 [2.93–5.46] | 3.88 [2.51–6.01] | |
| Iraq | 4.92 [3.65–6.62] | 3.63 [2.78–4.74] | | 5.87 [3.69–9.34] | 3.54 [2.20–5.71] | |
| Europe | 5.06 [3.74–6.85] | 5.43 [3.78–7.79] | | 6.68 [4.25–10.50] | 7.93 [4.51–13.96] | |
| **Level of education** | | | | | | |
| 0–9 years | 1.18 [1.01–1.38] | 1.28 [1.12–1.47] | df2, 2.10 (0.35) | 1.18 [0.90–1.54] | 1.50 [1.17–1.92] | df2, 1.75 (0.42) |
| 10–12 years | 1.11 [0.94–1.31] | 1.06 [0.91–1.23] | | 1.05 [0.79–1.40] | 1.19 [0.91–1.56] | |
| >12 years | 1 | 1 | | 1 | 1 | |
| **Children in the household** | | | | | | |
| Yes | 1 | 1 | df1, 8.16 (0.04) | 1 | 1 | df1, 2.23 (0.14) |
| No | 1.58 [1.41–1.77] | 1.28 [1.15–1.41] | | 2.09 [1.70–2.57] | 1.701 [1.408–2.055] | |
| **Job** | | | | | | |
| Yes | 1 | 1 | df1, 0.46 (0.50) | 1 | 1 | df1, 0.24 (0.63) |
| No | 1.62 [1.47–1.8] | 1.70 [1.55–1.86] | | 1.95 [1.61–2.35] | 2.08 [1.73–2.49] | |
| **Disposable income** | | | | | | |
| Reasonable living standard (≥100%) | 1.00 [0.92–1.09] | 1.14 [1.05–1.24] | df1, 4.93 (0.03) | 0.95 [0.81–1.11] | 1.33 [1.14–1.56] | df1, 9.32 (<0.01) |
| Below reasonable living standard (< 100%) | 1 | 1 | | 1 | 1 | |
| **Urbanicity** | | | | | | |
| Major city | 1.14 [0.97–1.35] | 1.61 [1.36–1.91] | df2, 9.5 (<0.01) | 1.14 [0.85–1.53] | 1.49 [1.07–2.07] | df2, 1.68 (0.43) |
| Medium-sized city | 0.97 [0.85–1.10] | 1.07 [0.90–1.28] | | 0.92 [0.73–1.16] | 1.17 [0.84–1.63] | |
| Smaller city or rural area | 1 | 1 | | 1 | 1 | |

*Not Somalia,

**Not Iraq.

in institutional housing during the asylum process have a particularly negative impact on mental health. These studies have focused on the associations between mental health and overcrowding and a lack of privacy, limited autonomy, and isolation from the surrounding community, all of which contribute to worsening symptoms of mental ill-health [19,20,29]. In addition, in a study comparing different asylum support systems and their effects on mental health it was shown that residents of centralized accommodations who were isolated from the population had worse mental health than did those at accommodations placed in the community [11]. Moreover, regarding linking different health outcomes with various housing forms, there are also studies that distinguish private from institutional housing, showing that institutional housing is associated with increased stress and a poorer quality of life among asylum seekers [12,13]. Similar to these studies, our findings indicate that institutional accommodations during the asylum process may be linked to more detrimental effects on mental health compared to private accommodations. What our study adds is that this association may extend to long-term mental health outcomes as well.

When it comes to the generalizability of our results, it is challenging to determine their applicability to other contexts. However, since refugees often share a pre-migration context, and assuming that the post-migration context does not significantly differ in policies and practices —an assumption that is often difficult to defend, given the substantial variation that typically exists between countries— the results may still provide some valuable insights for other contexts. Nonetheless, there is significant mobility in the policy area of asylum reception, and asylum policies within the EU are rapidly changing, affecting how valid over time these results may be.

While our findings support the importance of housing type for mental health, further attention to the mechanisms and contextual factors underlying this association would be beneficial. In the German context, Mohsenpour et al. [30] have devised a typology that distinguishes between varying refugee accommodations in order to deepen the understanding behind different mechanisms explaining the association between housing and mental health. They clustered refugee accommodations according to number of inhabitants, level of housing deterioration, urbanity of location, and remoteness to essential services, concluding that accommodations that were less crowded, had the lowest levels of housing deterioration, and were centrally located or urban had the best health outcomes in terms of CMDs. In the Swedish context, the categorization of ABO entails substantial variability and contextual differences when it comes to number of inhabitants and the location and quality of housing. For example, about half of the accommodations are apartments within the community. Single individuals share these apartments with unrelated others, while those who have families share them only with their own relatives. The other half comprises collective accommodations that vary in size and location. Considering this, one can argue that with more differentiated data on environmental and contextual conditions the analysis in the current study could have provided deeper insights and more accurate predictions. However, we did not have access to this specific data. Based on the available information, it is reasonable to state that the common denominator for ABO (institutional accommodations in the Swedish context) is that they are assigned following a dispersal and no-choice policy. It is important to consider this distinction when interpreting the current study's findings. The dispersal and no-choice policy is closely related to psychosocial factors, such as lack of control and autonomy [10], which we argue might constitute the most notable difference between private and institutional accommodations in Sweden. This is further supported by the study conducted with asylum seekers in EBO, which highlights that a significant factor in opting against ABO is the perceived loss of autonomy and control associated with not having the freedom to choose their living arrangements [21]. However, we encourage future research to go beyond the distinction between private and institutional housing to explore specific contextual factors and investigate the links to mental health in greater detail.

Taking advantage of nationwide registers in Sweden, we used data on healthcare utilization as a comparative measurement among similar migrant groups. However, to compare our two study groups and understand how the estimated differences might be explained, we need to consider whether they significantly differ in factors influencing the use of healthcare services. To conceptualize health service use in general, Andersen established a behavioral model encompassing three components: predisposition, enablement, and need [31,32]. The predisposing component includes demographic factors

such as age, sex, nationality, and level of education, whereas the enabling component includes characteristics of the healthcare system, social support, and factors such as income and health insurance [33]. The need component assumes that the individual must perceive illness for the utilization of health services. When it comes to predisposition, our study groups differed in several sociodemographic characteristics (see Table 1). However, as we adjusted in our analysis for sociodemographic factors such as age, sex, region of origin, children and level of education, which did not change our estimates, it is reasonable to assume that predisposing components have been accounted for to a considerable extent (Model 1, Table 3). When it comes to enablement—particularly the characteristics of the healthcare system—the Swedish healthcare system, with its universal coverage and relatively low out-of-pocket expenses, suggests that care is available and accessible in a similar manner for both groups, both legally and economically. However, it is conceivable that regional disparities and urban-rural differences in the availability of healthcare services exist in a vast country like Sweden, and that this availability also depends on transportation possibilities. Nevertheless, we argue that we have accounted for this by controlling for the level of urbanicity in our analysis. Enablement also entails social support and social networks as predictors of using healthcare services, with evidence showing that higher levels of social support are connected to increased use of mental health services [34]. Although our data lacks information on social support or networks, findings from other studies suggest that community housing promotes social integration, implying likely differences in social support between our two study groups [11,12]. However, this would imply that people from EBO would be more likely to seek care than those from ABO, which is contrary to our findings. In summary, while healthcare utilization is not to be used as a proxy for mental health, we nonetheless believe that the difference observed in this study can primarily be attributed to the fact that CMDs are more common among asylum seekers who have previously resided in ABO as compared to EBO. In addition to recognizing social support and social networks as factors influencing healthcare utilization, it's essential to acknowledge their impact on overall mental health and well-being. Various studies emphasize that social support is a crucial enabler of mental health among migrant groups, while a lack of social networks and support can negatively affect mental health [4,35]. In the Swedish context, research on a comparable refugee population has underscored the importance of social support for general health [36], suggesting that variations in CMDs could partly stem from differences in available social support. Forced migration generally disrupts pre-existing networks and limits access to broader social ties [37,38], yet the assumption that EBO individuals may choose this option due to having family or friends in Sweden suggests they might have greater access to social resources, which thereby could have an influence on mental health. Furthermore, living with family and friends in the community may offer a greater social network and social support, potentially serving as a buffer against mental ill-health. Given this, accounting for social support could have enhanced the robustness of our findings. However, this data is not accessible in Swedish registries, highlighting an area for future research. We therefore encourage longitudinal survey studies to further explore the role of social support in mental health for a more comprehensive understanding of the associations between housing and different mental health outcomes.

## Strengths and limitations

Our study has several strengths, one of which is its use of high-quality Swedish register data [39]. The register data allowed us to follow an entire population of former asylum seekers on a wide range of variables over a period of five years; this design minimizes selection and recall biases and reduces non-response bias. The longitudinal design is also a strength due to its potential to offer deeper insights into causal relationships between housing type and long-term healthcare use. It is also a strength to have two different outcome measures that showed consistency in their relationships with the exposure. Conducting additional sensitive analysis to account for the influence of missing data is another strength.

However, we also acknowledge several limitations to our study. An inherent limitation is its focus solely on healthcare utilization, without directly reflecting the true health status or overall well-being of the individual. Thus, it is essential to interpret the findings cautiously, considering migrant-specific barriers to healthcare. Furthermore, although there is high accuracy in the diagnoses in Swedish registers [40], the National Patient Register only contains information on specialized

health services and not from primary care. This means that we might have overlooked individuals presenting with milder symptoms and who were never referred to specialist care. However, this might partly be compensated for by the data on prescribed medications, which also includes prescriptions from primary healthcare centers. Nevertheless, it may still imply that individuals with milder symptoms of mental illness who seek healthcare but do not require antidepressant or anti-anxiety medication have been missed in this study. Another notable limitation is that up to 10% of the study population may have experienced both exposures, moving between ABO and EBO. While it is difficult to determine the reasons behind this changing of housing type and its potential association with mental health, it is important to interpret our findings with this consideration in mind.

In this study we controlled for potential confounding factors. However, we cannot rule out the possibility of residual confounding by other significant factors relevant to this context. For example, several other studies have highlighted aspects such as the length of the asylum process, language proficiency, and pre-migration trauma as important factors affecting the mental health of refugees and asylum seekers [4,41]. While these are important factors, we have no reason to believe they would vary differentially based on housing type, and no studies support this notion. Nonetheless, it would have further strengthened the validity of our study if we also had adjusted for these factors. Furthermore, as already mentioned, another source for residual confounding could be social support or social networks. In relation to our study, it is reasonable to consider that EBO and ABO might differ in terms of social support as outlined earlier. Accounting for the level of social support might have added depth to our analysis and explained part of the association between housing type and mental health service use. Furthermore, we did not have access to information on marital status, and consequently did not adjust for this in our analysis. This could be a limitation, given that marital status can be associated with healthcare utilization [42]. However, we adjusted for living with children and instances of divorce or widowhood, which we believe somewhat compensates for the lack of data on marital status. In adjusting for potential mediators, we used disposable income as the sole variable for income status; this might be a limitation, as disposable income does not distinguish between different sources of income such as welfare benefits or wages, which is essential for understanding economic status in a more nuanced way. Additionally, disposable income is calculated based on the family's total income, offering no insight into an individual's access to or needs for resources. Our mediators yielded no influence on the associations, which may be due to the use of variables with limited measurement precision such as disposable income, or missing factors such as social support. More research is warranted in order to deepen the understanding in this area.

Another limitation of our study is that it only addresses the long-term effects of housing on the mental health of those whose asylum applications were accepted. During the period in question, nearly 60% of applicants had their asylum applications rejected [14], and the long-term effects of accommodations type on the mental health of this group are not reflected in the study's results. An alternative that would have included all asylum seekers could have been to investigate differences in mental health service use during the actual time spent as an asylum seeker, but as national registers in Sweden do not contain information on people who are asylum seekers this would have meant a completely different study in scope and design. Nevertheless, we encourage future research to investigate this area in more detail.

## Conclusions

This study underlines that housing during the asylum process has an essential role in shaping the long-term mental health of refugees. We estimated the association between housing type during the asylum period and mental healthcare utilization up to five years after being granted a residence permit in Sweden. Although we accounted for sociodemographic factors known to affect the use of mental health services, there remained an unexplained association at the accommodations level. This indicates that individuals who lived in institutional housing (ABO) during their asylum period had an increased risk of using antidepressants or anxiolytics, as well as a higher likelihood of in- or outpatient visits in specialized care with a common mental disorder (CMD) diagnosis. The results align with previous research and underscore the need for closer monitoring and improvement of the housing situation among asylum seekers. This is particularly relevant at a time when

the reception of asylum seekers increasingly involves their living in large, collective facilities during the asylum period. When accommodations are assigned based on dispersal and non-choice policies, it is especially important to also discuss how mental health can be promoted during the asylum-seeking period, including the implementation of targeted mental health support interventions that address the unique challenges faced by individuals in these situations. Additionally, involving asylum seekers in the discussion of mental health promotion strategies is crucial to ensure that their needs and experiences are considered. To advance these discussions, future research should aim to develop a more comprehensive understanding of the links between housing and mental health for asylum seekers, with a particular focus on contextual factors and the role of social support. Such research should include both qualitative studies that provide in-depth insights into the lived experiences and challenges of asylum seekers as well as quantitative studies – ideally longitudinal surveys – to monitor the health outcomes over time and comparing the effects of different housing types. Ideally, these studies would incorporate a broad set of measures, including social support, pre-migration trauma, mental health status, and the duration of the asylum process. Our study has demonstrated that the mental health effects can persist long after an individual has ceased to be an asylum seeker.

## Supporting information

**S1 Data. GEE models.**
(XLSX)

## Acknowledgments

We would like to express our gratitude to Richard Bränström for facilitating access to essential register data, making this research possible.

## Author contributions

**Conceptualization:** Charlotta van Eggermont Arwidson, Petter Tinghög, Kristina Gottberg, Jessica Holmgren.

**Data curation:** Charlotta van Eggermont Arwidson.

**Formal analysis:** Charlotta van Eggermont Arwidson, Petter Tinghög.

**Methodology:** Charlotta van Eggermont Arwidson, Petter Tinghög, Kristina Gottberg.

**Project administration:** Charlotta van Eggermont Arwidson.

**Writing – original draft:** Charlotta van Eggermont Arwidson.

**Writing – review & editing:** Charlotta van Eggermont Arwidson, Petter Tinghög, Kristina Gottberg, Jessica Holmgren.

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
