## [Decision Letter · Decision Letter 0]

18 Oct 2024

PGPH-D-24-01909

Housing during the asylum process and its association with healthcare utilization for common mental disorders among refugees in Sweden: A nationwide cohort study

Dear Dr. van Eggermont Arwidson,

Thank you for submitting your manuscript to PLOS Global Public Health. After careful consideration, we feel that it has merit but does not fully meet PLOS Global Public Health’s publication criteria as it currently stands. Therefore, we invite you to submit a revised version of the manuscript that addresses the points raised during the review process.

We look forward to receiving your revised manuscript.

Kind regards,

Ryan Essex

Academic Editor

Journal Requirements:

 1. Please amend your detailed Financial Disclosure statement. This is published with the article. It must therefore be completed in full sentences and contain the exact wording you wish to be published. **Please only choose the relevant sentences from below** 1. Please clarify all sources of funding (financial or material support) for your study. List the grants (with grant number) or organizations (with url) that supported your study, including funding received from your institution. 2. State the initials, alongside each funding source, of each author to receive each grant.3. State what role the funders took in the study. If the funders had no role in your study, please state: “The funders had no role in study design, data collection and analysis, decision to publish, or preparation of the manuscript.”4. If any authors received a salary from any of your funders, please state which authors and which funders. If you did not receive any funding for this study, please simply state: “The authors received no specific funding for this work.” 2. In the online submission form, you indicated that "The register data used in this study contain sensitive information and under swedish law and the ethical approval, patient-level data cannot be publicly available. Codes can be made available upon request.".  All PLOS journals now require all data underlying the findings described in their manuscript to be freely available to other researchers, either 1. In a public repository, 2. Within the manuscript itself, or 3. Uploaded as supplementary information. This policy applies to all data except where public deposition would breach compliance with the protocol approved by your research ethics board. If your data cannot be made publicly available for ethical or legal reasons (e.g., public availability would compromise patient privacy), please explain your reasons by return email and your exemption request will be escalated to the editor for approval. Your exemption request will be handled independently and will not hold up the peer review process, but will need to be resolved should your manuscript be accepted for publication. One of the Editorial team will then be in touch if there are any issues.

Reviewers' comments:

Reviewer's Responses to Questions

**Comments to the Author**

1. Does this manuscript meet PLOS Global Public Health’s publication criteria ? Is the manuscript technically sound, and do the data support the conclusions? The manuscript must describe methodologically and ethically rigorous research with conclusions that are appropriately drawn based on the data presented.

Reviewer #1: Yes

Reviewer #2: Yes

2. Has the statistical analysis been performed appropriately and rigorously?

Reviewer #1: Yes

Reviewer #2: I don't know

3. Have the authors made all data underlying the findings in their manuscript fully available (please refer to the Data Availability Statement at the start of the manuscript PDF file)?

Reviewer #1: No

Reviewer #2: Yes

4. Is the manuscript presented in an intelligible fashion and written in standard English?

Reviewer #1: Yes

Reviewer #2: Yes

5. Review Comments to the Author

Reviewer #1: This is a very important contribution to existing literature. Thank you. Your analysis and/or discussion could be improved by few factors that were not covered in the analysis. Please see the attached document.

Reviewer #2: The authors have presented the results of a nationwide cohort study on housing during the asylum process and its association with healthcare utilization for common mental disorders among refugees in Sweden. The manuscript is well written and addresses an important public health issue, particularly given that housing in the post-migration context plays a crucial role in shaping the mental well-being of refugees and asylum seekers. However, there are a few areas where the paper could be strengthened to enhance its contribution to the field, please find specific comments below:

Introduction:

The introduction provides a strong rationale for the current study, but additional context on how this research builds on and differentiates from existing literature on CMDs among refugees, particularly in the Swedish setting, could be beneficial.

As this study has significant policy implications, adding a brief statement about the current debate around EBO (self-organized housing) and ABO (institutional housing) models could strengthen the introduction and highlight the potential impact of the findings.

The introduction touches on housing’s role in mental health. Adding a few lines on the pathways linking housing and mental health—such as housing affordability, tenure stability and housing satisfaction—could clarify the conceptual basis for the study.

Methods:

Line 166: South America was coded as missing (n=20), this needs to be highlighted in the limitations. Was any alternative approach considered?

A detailed explanation of why certain sociodemographic factors were considered mediators rather than confounders would clarify the modelling strategy. It would also be helpful to note how missing data were handled for the time-dependent covariates.

Results:

This section is well presented.

Discussion:

The policy implications could be expanded. Given that Swedish asylum policy is expected to shift towards more institutionalized accommodation, the authors could discuss how this research might inform future housing policy and mental health support interventions.

Although the manuscript addresses several limitations, an additional note on the potential residual confounding effects of pre-migration trauma, social support, and length of stay in Sweden would provide a more comprehensive view. Expanding on the impact of not including social support measures as potential confounders would be useful, as differences in social support could be significant between ABO and EBO groups.

The call for future research could be strengthened by specifying particular areas to explore, such as longitudinal studies that follow asylum seekers in both housing types from arrival through post-asylum periods, or qualitative studies to capture personal experiences within ABO and EBO settings.

6. PLOS authors have the option to publish the peer review history of their article (what does this mean? ). If published, this will include your full peer review and any attached files.

**Do you want your identity to be public for this peer review?** For information about this choice, including consent withdrawal, please see our Privacy Policy .

Reviewer #1: No

Reviewer #2: **Yes: ** Kritika Rana

---

## [Decision Letter · Decision Letter 1]

29 Jan 2025

PGPH-D-24-01909R1

Housing during the asylum process and its association with healthcare utilization for common mental disorders among refugees in Sweden: A nationwide cohort study

Dear Dr. van Eggermont Arwidson,

Thank you for your responses to our previous requests. I apologize that we are requesting further updates, but it is PLOS Global Public Health's goal to provide the highest standard of reporting in manuscripts that are published with us and there are a few more points we would like to clarify before we can proceed. Therefore, we invite you to submit a revised version of the manuscript that addresses the points raised during the review process.

Upon further review we are now in a position to grant an exemption on sharing your data as per the journal's requirements. However, even with a data sharing exemption we require additional information to be provided in your Data Availability Statement (DAS) before your manuscript can proceed to publication. Please can you update your DAS ensuring that it includes the following information:

a) Discuss what data you are providing with the paper and the reason why the full underlying data set cannot be made available to other researchers, even upon request.

b) List multiple authors as points of contact and include their contact information (i.e., email address).

c) Provide additional details on how you will ensure persistent or long-term data storage and availability.

d) Specify in your Data Availability statement who will have access to the data and the criteria you will use to consider requests (e.g., the limitations of data sharing as outlined in the IRB approved protocol or patient consent form).

e) Include the following text in your Data Availability statement: “Although the authors cannot make their study’s data publicly available at the time of publication, all authors commit to make the data underlying the findings described in this study fully available without restriction to those who request the data, in compliance with the PLOS Data Availability policy. For data sets involving personally identifiable information or other sensitive data, data sharing is contingent on the data being handled appropriately by the data requester and in accordance with all applicable local requirements.

We look forward to receiving your revised manuscript.

Kind regards,

Emma Campbell, Ph.D

Staff Editor

Journal Requirements:

Reviewers' comments:

Reviewer's Responses to Questions

**Comments to the Author**

1. If the authors have adequately addressed your comments raised in a previous round of review and you feel that this manuscript is now acceptable for publication, you may indicate that here to bypass the “Comments to the Author” section, enter your conflict of interest statement in the “Confidential to Editor” section, and submit your "Accept" recommendation.

Reviewer #2: All comments have been addressed

2. Does this manuscript meet PLOS Global Public Health’s publication criteria ? Is the manuscript technically sound, and do the data support the conclusions? The manuscript must describe methodologically and ethically rigorous research with conclusions that are appropriately drawn based on the data presented.

Reviewer #2: Yes

3. Has the statistical analysis been performed appropriately and rigorously?

Reviewer #2: I don't know

4. Have the authors made all data underlying the findings in their manuscript fully available (please refer to the Data Availability Statement at the start of the manuscript PDF file)?

Reviewer #2: Yes

5. Is the manuscript presented in an intelligible fashion and written in standard English?

Reviewer #2: Yes

6. Review Comments to the Author

Reviewer #2: (No Response)

7. PLOS authors have the option to publish the peer review history of their article (what does this mean? ). If published, this will include your full peer review and any attached files.

**Do you want your identity to be public for this peer review?** For information about this choice, including consent withdrawal, please see our Privacy Policy .

Reviewer #2: **Yes: ** Kritika Rana

---

## [Editor Report · Decision Letter 2]

7 Apr 2025

Housing during the asylum process and its association with healthcare utilization for common mental disorders among refugees in Sweden: A nationwide cohort study

PGPH-D-24-01909R2

Dear PhD student van Eggermont Arwidson,

We are pleased to inform you that your manuscript 'Housing during the asylum process and its association with healthcare utilization for common mental disorders among refugees in Sweden: A nationwide cohort study' has been provisionally accepted for publication in PLOS Global Public Health.

Best regards,

Ryan Essex

Academic Editor